# Could IL-17A Be a Novel Therapeutic Target in Diabetic Nephropathy?

**DOI:** 10.3390/jcm9010272

**Published:** 2020-01-19

**Authors:** Carolina Lavoz, Sandra Rayego-Mateos, Macarena Orejudo, Lucas Opazo-Ríos, Vanessa Marchant, Laura Marquez-Exposito, Antonio Tejera-Muñoz, Juan F. Navarro-González, Alejandra Droguett, Alberto Ortiz, Jesús Egido, Sergio Mezzano, Raúl R. Rodrigues-Diez, Marta Ruiz-Ortega

**Affiliations:** 1Laboratorio de Nefrología, Facultad de Medicina, Universidad Austral de Chile, Valdivia 5090000, Chile; carolavoz@gmail.com (C.L.); vmarchant.hernandez@gmail.com (V.M.); m.aledroguett@gmail.com (A.D.); mezzano.sergioa@gmail.com (S.M.); 2Vascular and Renal Translational Research Group, Institut de Recerca Biomèdica de Lleida (IRBLleida), 25198 Lleida, Spain; srayego@fjd.es; 3Red de Investigación Renal (REDINREN), Instituto de Salud Carlos III, 28029 Madrid, Spain; macarena.orejudo@quironsalud.es (M.O.); laura.marqueze@quironsalud.es (L.M.-E.); antoniotemu@gmail.com (A.T.-M.); aortiz@fjd.es (A.O.);; 4Cellular and Molecular Biology in Renal and Vascular Pathology Laboratory, Fundación Instituto de Investigación Sanitaria-Fundación Jiménez Díaz-Universidad Autónoma Madrid, 28040 Madrid, Spain; 5Renal, Vascular and Diabetes Research Laboratory, Fundación Instituto de Investigación Sanitaria-Fundación Jiménez Díaz-Universidad Autónoma Madrid, 28040 Madrid, Spain; lucasopazo78@gmail.com (L.O.-R.); jegido@fjd.es (J.E.); 6Spanish Biomedical Research Centre in Diabetes and Associated Metabolic Disorders (CIBERDEM), Instituto de Salud Carlos III, 28029 Madrid, Spain; 7Unidad de Investigación y Servicio de Nefrología, Hospital Universitario Nuestra Señora de Candelaria, 38010 Santa Cruz de Tenerife, Spain; jnavgon@gobiernodecanarias.org; 8Nephrology and Hypertension, Fundación Instituto de Investigación Sanitaria-Fundación Jiménez Díaz-Universidad Autónoma Madrid, 28040 Madrid, Spain

**Keywords:** diabetic nephropathy, inflammation, immune cells, cytokines, IL-17A, treatment, diabetes mellitus, proteinuria

## Abstract

Chronic kidney disease has become a major medical issue in recent years due to its high prevalence worldwide, its association with premature mortality, and its social and economic implications. A number of patients gradually progress to end-stage renal disease (ESRD), requiring then dialysis and kidney transplantation. Currently, approximately 40% of patients with diabetes develop kidney disease, making it the most prevalent cause of ESRD. Thus, more effective therapies for diabetic nephropathy are needed. In preclinical studies of diabetes, anti-inflammatory therapeutic strategies have been used to protect the kidneys. Recent evidence supports that immune cells play an active role in the pathogenesis of diabetic nephropathy. Th17 immune cells and their effector cytokine IL-17A have recently emerged as promising targets in several clinical conditions, including renal diseases. Here, we review current knowledge regarding the involvement of Th17/IL-17A in the genesis of diabetic renal injury, as well as the rationale behind targeting IL-17A as an additional therapy in patients with diabetic nephropathy.

## 1. Introduction

Diabetic nephropathy (DN) is a significant microvascular complication of diabetes. The epidemic of type 2 diabetes is the main cause of chronic kidney disease (CKD) worldwide, which leads to premature death and end-stage renal disease (ESRD). Up to a third of patients with type 1 and type 2 diabetes are estimated to develop DN. In the absence of DN, mortality among diabetic patients is comparable to that of the general population [1,2]. CKD is expected to become the fifth global cause of death by 2040, and the second in long lived countries before the end of the century, mainly driven by DN [3,4]. In this review, we discuss the novel findings that support the contribution of an inflammatory microenvironment to the pathogenesis of DN. The potential for anti-inflammatory therapeutic strategies is also discussed, remarking about the importance of Th17 mediated immune response and its effector cytokine, interleukin 17A (IL-17A).

## 2. Current Treatments of Diabetic Nephropathy

At present, the core treatment of DN relies on an optimal control of the renin–angiotensin–aldosterone (RAAS) system using angiotensin converting enzyme inhibitors (ACEI), angiotensin receptor blockers (ARB), or aldosterone blockers (spironolactone or finerenone) [5]. Maximal RAAS blockade strategies, such as dual blockade approaches (ACEI plus ARB or renin inhibitor), have had disappointing results in lowering the risk of albuminuria and reducing the risk of ESRD and have increased the risk of adverse events [6]. Although intensive glycemic control has been shown to delay the onset and progression of DN, it poses challenges due to the high risk of hypoglycemia and alterations in the pharmacokinetics of anti-hyperglycemic drugs [7,8]. However, recent and growing evidence has shown the cardiovascular and renal safety and efficacy of newer antihyperglycemic medications, such as DPP-4 (dipeptidyl peptidase-4) inhibitors, GLP-1 RA (glucagon-like peptide-1 receptor agonist), and SGLT-2 (sodium-glucose cotransporter 2) inhibitors [9]. The combination of RAAS blockade with SGLT2 inhibitors and/or GLP-1 RA represents the new standard for kidney and heart protection in DN [10]. Indeed, SGLT2 inhibitors have demonstrated to be the first drugs to decrease both cardiovascular and renal events in DN patients in over 20 years [11]. 

Due to the complex mechanisms involved in the onset of renal injury in diabetes, it is highly unlikely that new anti-diabetic drugs, even when used appropriately and rationally, would halt the progression of DN, as evidenced by the persisting residual risk. Among the different pathways dysregulated by chronic hyperglycemia, inflammation plays a key and predominant role. Initial studies on streptozotocin (STZ)-induced diabetes in rats first proposed that inflammatory factors contribute to the pathogenesis of DN [12]. Since then, the key role of inflammation in the pathophysiology process of experimental and human DN has been demonstrated. The molecular mechanisms involved have also been elucidated, including oxidative stress [13,14], activation of nuclear factor kappa-B (NF-κB) [14], and production of related cytokines such as tumor necrosis factor (TNF) [13,14,15] and Toll-like receptor (TLR) activation [16]. Accordingly, targeting of macrophages has been extensively studied as an anti-inflammatory strategy [17]. In a recent review, we provided thorough information on the experimental and clinical studies reporting the beneficial effects of agents targeting inflammation pathways, such as monocyte protein-1 (MCP-1/CCL2), its receptor CCR2 (chemokine ligand 2/C-C-motif chemokine receptor type 2), IL-1β, and JAK/STAT (Janus kinase/signal transducer and activator of transcription), as well as Nrf2 (nuclear factor erythroid 2 related factor 2) inducers, e.g., bardoxolone methyl [18]. Use of bardoxolone methyl in clinical trials, such as the BEACON CRC study (Identifier: NCT02928224), did not lead to successful results [19,20]. Despite this, new therapies targeting intracellular inflammatory signaling pathways are emerging, such as the selective JAK inhibitor baricitinib and the inhibitor of apoptosis signal regulating kinase 1 (ASK-1) selonsertib [21,22].

## 3. Biomarkers of Diabetic Nephropathy

The growing global impact of DN, together with the recent ongoing clinical trials on novel therapeutic approaches, have increased the need for novel biomarkers that allow an earlier diagnosis of renal involvement. These tools could let early intervention or prediction of therapy response, leading to a personalized therapeutic approach [3,4,23,24]. One of these tools is urine proteomics, which is a promising strategy for predicting rapid disease progression before albuminuria becomes pathological [25]. Recently, a kidney risk inflammatory signature consisting of 17 circulating proteins was associated with the 10-year risk of ESRD in type 1 and type 2 DN with pathological albuminuria and low estimated glomerular filtration rate [26]. These proteins were not thought to be of kidney origin and included six TNF superfamily receptors and additional cytokines, such as IL-17F, cytokine receptors, and chemokines. These biomarkers were associated with ESRD both in an albuminuria dependent and independent manner, suggesting the activation of several pathways that promote kidney injury. Interestingly, kidney risk inflammatory signature proteins were not responsive to RAAS blockade, but decreased with the use of JAK1/2 inhibitor baricitinib. This suggests the existence of a kidney injury pathway resistant to current nephroprotective strategies, and thus, there may be patients who might benefit from anti-inflammatory therapy [26].

## 4. Immune Cells in the Pathogenesis of Diabetes

Immune cells play a key role in the pathogenesis of immune and chronic inflammatory diseases. After antigenic stimulation, naive CD4+ T lymphocytes are activated and differentiate into several T helper (Th) effector subpopulations [27], such as Th1, Th2, and Th17 cells [28]. Differentiation of CD4 cells into the diverse cell subtypes is tightly regulated by specific transcription factors and cytokines [29]. Th17 differentiation is controlled by the transcription factor retinoid related orphan receptor γt (RORγt) and the activation of STAT-3 (Figure 1) [30]. Each Th subtype produces a specific cytokine pattern. Th17 cells produce cytokines of the IL-17 family, IL-17A being the main effector, which increase the expression of CCR6 [31]. The IL-17 family of proteins binds to specific receptors (IL-17RA–IL-17RE), activating downstream signaling systems such as the NF-κB pathway and redox mechanisms [32,33].

### 4.1. Th17 Immune Cells and IL-17A in Human Diabetes and Diabetic Nephropathy

Emerging evidence suggests that immune cells may play an active role in the pathogenesis of diabetes. When focusing on Th17 cells and human diabetes, there is evidence for an altered homeostasis. In type 1 diabetic patients, there is an imbalance in the ratio of circulating Treg/Th17 [34]. In addition, in long term type 1 diabetic mice that still present residual pancreatic β-cell function, the number of circulating IL-17A+ cells was higher than the number of Tregs, CD4+ T cells, and CD8+ T cells [35]. Moreover, serum levels of several Th17 related cytokines, including IL-17A and IL-21, were higher in diabetic patients than in controls [36,37]. In type 2 diabetic patients, there is a low grade systemic inflammation associated with altered T cell populations, including reduced overall T cells, Th17, IL-21R+, Tregs, and TLR4+ T cells, while monocytes show enhanced TLR4 expression; however, the Treg/Th17 ratio was not explored [38]. Moreover, immune cells from these patients presented altered mitochondrial function and compromised β oxidation [39], showing a metabolic reprogramming in the Th17 immune cells in diabetes. 

### 4.2. IL-17 in Human Diabetic Nephropathy

IL-17F has been described as a circulating inflammatory protein associated with increased risk of renal damage progression [26]. Interestingly, circulating IL-17A levels are related to the severity of kidney disease and progressively decrease from subjects with normal glucose tolerance to subjects with type 2 diabetes with and without DN [40]. In agreement with this, patients with advanced DN present lower levels of IL-17A in both plasma and urine [41]. However, Zhang et al. showed an increase in CD4+ CXCR5+ PD-1+ T follicular helper cells and plasma values of IL-6 and IL-17 in patients with DN compared to healthy controls [42]. Among cirrhotic hepatitis C virus infected patients, serum IL-17A levels were higher in those that were type 2 diabetic than in non-diabetic patients and controls [43]. Although these studies have addressed circulating or urinary IL-17A levels in DN patients, local renal levels of IL-17A have not been investigated yet. Importantly, infiltration of immune cells is a key feature of DN [17]. Activated T cells (CD4+ and CD8+) are mainly located in the renal interstitium of diabetic kidneys [44,45,46]. Although CD4+ IL-17+ cells are the main source of IL-17A production, other cells, including macrophages, neutrophils, natural killer, dendritic, and mast cells, have also been described to produce this cytokine. All these cells release proinflammatory and profibrotic factors such as CD40L, IL-6, transforming growth factor-β1 (TGF-β1), Rantes, and MCP-1, which act synergistically in the progression of DN [47]. In this sense, in a pioneer study in renal biopsies of DN patients, we described the local activation of inflammatory pathways, specifically NF-κB activation linked to an upregulation of proinflammatory factors, such as the chemokine MCP-1 [14]. Since then, many preclinical studies have demonstrated that MCP-1 can be a therapeutic target and potential biomarker for DN. In this regard, clinical trials targeting MCP-1 or its receptor have shown promising results [48]. Nevertheless, studies evaluating the kidney expression of IL-17A in human DN are needed to further define its role in DN progression.

### 4.3. Role of Th17 Immune Cells and IL-17A in the Development of Experimental Diabetes

Several preclinical studies have confirmed the contribution of Th17 cells and IL-17A to diabetes and diabetes end-organ damage, pointing to Th17/IL17A blockade as beneficial in diabetes. The earliest experimental studies were carried out in BDC2.5 T cell receptor transgenic non-obese diabetic (NOD) mice, which spontaneously develop destructive autoimmune insulitis and progress to overt diabetes, representing a convenient experimental model to study human type I DN [49]. In NOD mice, CD4+ T cells are activated and migrate into Langerhans islets to release inflammatory cytokines, such as interferon (IFN)-γ and IL-2, thereby recruiting inflammatory cells. Recruited cytotoxic macrophages and CD8+ T cells destroy pancreatic insulin producing β-cells through perforin/granzyme mediated toxicity in a process called insulitis [50]. In NOD mice, serum IL-17A levels and the number of pancreatic IL-17A producing Th17 cells and IFN-γ producing Th1 cells increased, thus identifying a mechanism by which Th17 cells can contribute to type 1 diabetes [35]. Naive CD4+ T lymphocytes from NOD mice can be polarized to Th17 by incubation with IL-23 plus IL-6. These Th17 cells release IL-17A and IL-22 and induce type 1 diabetes in young NOD mice upon adoptive transfer [51]. Moreover, IL-22 producing Th17 cells were also isolated from the pancreas of diabetic NOD mice [52]. 

Some preclinical studies have demonstrated the beneficial effects of Th17/IL-17A blockade in diabetes. Treatment of NOD mice with a selective inverse agonist of RORα/γ, the main transcription factor involved in Th17 differentiation, significantly reduced diabetes incidence, insulitis, and proinflammatory cytokine expression [53], showing that Th17 differentiation blockade improves experimental diabetes. Moreover, treatment with neutralizing anti-IL-17A in NOD mice prevented diabetes when treatment started at 10 weeks of age, but not at earlier stages [54]. Recently, the role of IL-17A in the development of insulitis was confirmed using an IL-17A/IFN-γ double deficiency receptor in NOD mice [55]. Accordingly, in IL-17A knockout mice with STZ induced diabetes, hyperglycemia and insulitis were milder than in wild-type mice [56]. On the other hand, Treg cells conferred a protective effect during the development of type 1 diabetes [50,57], and adoptive transfer of in vitro expanded antigen specific Treg cells prevented the development of diabetes and even restored an immune regulatory state that reversed it [58].

### 4.4. The Th17/IL-17A Axis in Experimental Diabetic Nephropathy

Many studies described the IL-23/IL-17A pathway as a novel therapeutic target against chronic inflammatory diseases, including renal disease with different etiologies [59,60,61]. In contrast, contradictory data have been described in experimental DN regarding the protective impact of IL-17A on the incidence of diabetes [62]. In IL-17A knockout mice with STZ induced diabetes, renal lesions were more severe in IL-17A deficient mice than in wild-type mice [41]. This result is striking, since IL-17A targeting had been reported to decrease the severity of diabetes itself. Moreover, in the same study administration of low dose of recombinant IL-17A or IL-17F reduced albuminuria and renal injury in Ins2 Akita mice, a model for type 1 diabetes [41]. These authors proposed that the protective effects of IL-17A and IL-17F could be attributed to inhibition of STAT-3 activation, but target cells were not identified [41]. In contrast, a recent study in STZ-diabetic model has described opposite results. A protective effect by treatment with a neutralizing IL-17A monoclonal antibody, and accordingly, the renal lesions were diminished in IL-17A deficient mice compared to wild-type mice [63], suggesting that IL-17A could promote DN. In fact, several other studies have demonstrated that elevated circulating IL-17A levels, achieved by gene overexpression or intraperitoneal or systemic administration of recombinant IL-17A, are deleterious for the vasculature and the kidney. Increased blood pressure, endothelial dysfunction, and inflammatory cell infiltration in the kidney are involved in this kind of damage [64,65]. Moreover, hypertensive patients have increased circulating IL-17A levels [66,67], suggesting that this cytokine can be involved in the onset of hypertension and hypertension related end-organ damage [68].

Another possible explanation for the discrepancy in the role of Th17/IL-17A in DN could be the limited translational value of DN animal models, due to the difficulties of translating experimental information that generally does not recapitulate the renal lesions observed in diabetic patients. Type 1 diabetes models, such as Akita mice and STZ administration in the C57BL/6 background, and type 2 models, such as db/db mice (leptin receptor deficient), are excellent examples for the study of earlier stages of DN, but they still lack translatability to the clinic [69]. Leptin deficient BTBR ob/ob mice are characterized by a kidney disease that mimics key features of advanced human DN with evidence of the reversibility of glomerular lesions. Due to this, this model is excellent for carrying out preclinical studies of therapeutic interventions [70,71]. Recently, we described that administration of a neutralizing anti-IL-17A antibody to BTBR ob/ob mice after kidney disease development reversed the structural abnormalities of DN, including amelioration of mesangial matrix accumulation, renal inflammation mitigation, and improved renal function [72], suggesting that IL-17A blockade could be a potential therapeutic option for DN. 

### 4.5. The Th17/IL-17A Axis in Diabetic Complications

Deregulated Th17 cells have been described in other diabetes related complications as well. A large body of evidence suggests an important role of Th17/IL-17A in diabetic retinopathy. Vitreous fluid IL-17A levels were higher in patients with proliferative diabetic retinopathy [73]. However, elevated circulating IL-17A levels or activated circulating immune cells were not found [73,74,75,76], suggesting local activation of the Th17 immune response, and not a systemic response, was involved in end-organ damage. Blocking IL-17A by intravitreal injections with neutralizing antibodies against IL-17A or its receptor slowed diabetic retinopathy progression by impairing retinal Müller cell function [75]. Additionally, in rats with STZ induced diabetic retinopathy, local injection of anti-IL-23 antibodies improved the blood–retinal barrier structure [76]. In obese diabetic mice, treatment with anti-IL-17A and anti-IL-23 antibodies improved wound re-epithelialization. In the same study, local wound IL-17A levels were lower in IL-23 deficient animals [77].

## 5. IL-17A as a Proinflammatory Mediator in DN

IL-17A responses vary depending on cell type and pathological conditions, exerting mainly proinflammatory responses [78,79,80,81,82]. In cultured cells, IL-17A regulates many proinflammatory factors, including chemokines, adhesion molecules, and cytokines [83,84,85,86]. In podocytes in vitro, IL-17-A increased IL-6 and TNF-α gene expression and showed additive effects on the regulation of these cytokines in the presence of high glucose [63]. In cultured tubular epithelial cells, stimulation with IL-17A also induces proinflammatory gene expression and increases MCP-1 production [63,83]. In these cells, IL-17A is also able to induce epithelial-to-mesenchymal transition (EMT) [87,88], including in cultured proximal tubular epithelial cells [89], and therefore, it might also contribute to renal damage by triggering this mechanism (Figure 2) [90]. Moreover, IL-17A activates other immune cells, including monocytes, by regulating chemotaxis [91,92] and contributing to their recruitment into injured tissues.

Many resident renal cells express receptors for IL-17A. After this ligand binds to IL-17RA/RC receptors, several intracellular signals can be activated (Figure 3). The main downstream mechanism involved in IL-17A signaling is the activation of the NF-κB pathway and downstream regulation of proinflammatory genes [33]. In leptin deficient BTBR ob/ob mice, the beneficial effects of a neutralizing anti-IL-17A antibody were associated with the inhibition of inflammation related pathways, including NF-κB activation and upregulation of related genes, such as MCP-1. This was independent of glycemic control [72]. Thus, IL-17A/NF-κB pathway activation contributes to renal inflammation under diabetic conditions. We have recently described that systemic administration of IL-17A in C57BL/6 mice significantly upregulated kidney *Mcp-1* and *Rantes* gene expression and recruited inflammatory cells to the kidney [64]. Moreover, in experimental angiotensin II induced renal damage, IL-17A neutralization also decreased proinflammatory genes and inflammatory cell infiltration [64,93]. These data suggest that the elevated local IL-17A production observed in diabetic kidneys could activate resident renal cells to produce proinflammatory cytokines and chemokines, such as MCP-1. This could contribute to the further recruitment of inflammatory cells into the diabetic kidney, amplifying the inflammatory response (Figure 3). The involvement of redox processes in IL-17A actions has also been described in endothelial and immune cells [66]. Another important signal activated by IL-17A includes the protein kinases, such as RhoA/Rho-kinase, MAPK cascade, and Akt signaling [33,64,66] (Figure 3).

## 6. Pharmacological Therapies Interfering with Th17 Immune Responses

Different anti-inflammatory strategies with beneficial effects in experimental diabetes may also improve T cell responses, including Th17 related effects [24]. In experimental STZ induced DN, mycophenolate mofetil diminished the number of CD4+/IL-17A+ cells in the kidney and suppressed renal T cell proliferation [94]. In human mononuclear cells in peripheral blood, sitagliptin, a DPP-4 inhibitor, diminished T cell proliferation and induced a Th cell phenotype switch to a Treg subtype with higher secretion of TGF-β1 and lower IL-17A gene expression [95]. In this regard, DPP-4 inhibitors improved β-cell function and attenuated autoimmunity in type 1 diabetic mice [24]. Immunotherapy with complete Freund’s adjuvant reduced the Th17 response and Th17 related cytokine levels in diabetic mice [96]. Treatment of NOD mice with metformin, an AMP activated protein kinase activator, reduced the severity of autoimmune insulitis by modulating the Th17/Treg balance [97]. The mechanism of action of metformin involves the inhibition of the mammalian target of rapamycin (mTOR), with the subsequent glycolysis inhibition and enhancement of lipid oxidation, which suggests that T cell metabolism could be a potential target for inhibiting Th17 differentiation and related deleterious effects.

## 7. MicroRNAs in Diabetic Nephropathy

MicroRNAs (miRNAs) are small single stranded non-coding RNAs [98]. They usually bind to the 3′ untranslated region of target mRNAs, leading to either degradation of the mRNA or to translational repression, finally diminishing the expression of the target gene [99,100] and, therefore, controlling gene expression [101]. There is strong evidence showing that aberrant miRNA expression can lead to the devolvement and progression of many pathophysiological processes, including cancer, diabetes, and cardiovascular diseases [102,103]. A wide range of miRNAs has been described to regulate glucose homeostasis and, therefore, the pathogenesis of diabetes. Several miRNAs regulate insulin. Insulin secretion is negatively regulated by overexpression of miR-375, miR-9, or miR-96 in β-cells [104]. Other miRNAs target insulin signaling, including miR-278, miR-14, and miR-29 in adipose tissue, miR-122 and miR-33 in liver, and miR-24 in skeletal muscle [104]. The identification of miRNAs as novel biomarkers for nephropathies, including DN, may contribute to more precise diagnosis and risk stratification, as well as provide valuable additional information for patient management, including miRNA targeting.

The activity of specific miRNAs in the kidney may be modulated by in vivo delivery of mimics that restore miRNA levels or inhibitors that block miRNA function. Successful kidney transfection has been achieved by intraperitoneal, intravenous, or subcutaneous injection of either mimics or inhibitors [105,106]. Therefore, miRNA regulation has been proposed as a promising therapeutic target for DN [107,108]. miR-146a deletion has been shown to accelerate DN development in mouse models [109]. Blocking the direct effects of miR-21 on podocytes in diabetic mice resulted in decreased podocyte loss, albuminuria, and interstitial fibrosis [110]. However, many inflammatory and profibrotic genes have been identified as targets of miR-21 [111]. Interestingly, in miR-21 knockout mice, renal damage was ameliorated, but most of the genes silenced after renal injury were involved in metabolic and mitochondrial functions, with peroxisome proliferator activated receptor-α (PPARα) being a direct target of miR-21 [110,112]. Other recent studies have described that miR-9 and miR-33 regulate metabolic pathways related to fatty acid oxidation, exerting protective effects in experimental renal damage [113,114]. Future preclinical studies are warranted for evaluating potential miRNAs as therapeutic targets. However, testing of the first miRNA targeted drug, miravirsen (anti-miR-122), raised a note of caution, since evidence of nephrotoxicity was observed during trials for hepatitis C virus, and clinical development appears to have stalled [115].

### miRNAs Involved in Th17 Differentiation

Differentiation into distinct T helper subtypes is tightly regulated to ensure an immunological balance. However, the precise regulatory networks of Th17 differentiation in complex diseases are still unknown, and their characterization may potentially allow developing novel therapies for Th17 related diseases. In different models of autoimmune diseases, several miRNAs were shown to regulate Th17 cell differentiation by targeting the transcription factors that activate Th17 differentiation or decrease Treg, including RORγT, STAT3, or forkhead box P3 (FOXP3), as well as key cytokines of this process, such as IL-21R (Table 1). 

There are few studies on the role of miRNAs in Th17 differentiation in kidney disease. A recent study described that miRNA-155 deficiency promotes nephrin acetylation and decreases renal damage in hyperglycemia induced nephropathy, effects that were associated with inhibited IL-17A production through enhancement of suppressor of cytokine signaling 1 (SOCS1) expression [116]. miRNA-155 is also overexpressed in the human anti-neutrophil cytoplasmic antibody (ANCA) associated crescentic glomerulonephritis. Additionally, in murine nephrotoxic nephritis, miR-155 knockout mice showed a significant reduction of the Th17 immune response, less severe nephritis, and reduced histologic and functional injury [117].

## 8. Is a Therapeutic Trial with Anti-IL-17 Antibodies in Diabetic Nephropathy Feasible?

IL-17A has emerged as an important inflammatory mediator involved in the genesis of immune and chronic inflammatory diseases, including cardiovascular and renal diseases [31,33,136], and diabetic complications, as described in this review. In this regard, there are several ongoing clinical trials using neutralizing antibodies against IL-17A for chronic human inflammatory diseases, such as chronic plaque psoriasis, psoriatic arthritis, ankylosing spondylitis, and rheumatoid arthritis [137,138]. Secukinumab, a fully human IgG1 kappa antibody, was the first anti-IL-17 biological agent to be approved by the U.S. Food & Drug Administration (FDA) and the European Medicines Agent (EMA) for the treatment of moderate-to-severe psoriasis and psoriatic arthritis in adult patients. Other IL-17A inhibitors already on the market include ixekizumab (a humanized anti-IL17 IgG4 monoclonal antibody) and brodalumab (a human IgG2 monoclonal antibody antagonizing the IL-17RA receptor). All anti-IL-17A agents were significantly more effective than anti-TNF alpha agents such as infliximab, adalimumab, and etanercept, but less so than certolizumab. Regarding safety, the anti-IL-17A antibodies are generally well tolerated and have a safety profile comparable to other antipsoriatic biologic agents [138]. However, the reported increased frequency of infections vs. placebo should be carefully monitored in an eventual diabetes trial.

In recent years, the field of anti-IL-17A antibodies in the treatment of human diseases has expanded considerably with new clinical trials and novel indications. At the time of writing this manuscript (December 2019), 62 studies on IL-17A neutralization were found in the online resource provided by U.S. National Library of Medicine (clinicaltrials.gov) related to established indications (psoriatic arthritis, rheumatoid arthritis with inadequate response to anti-TNF agents), but also to other indications, such as pyoderma gangrenosum, moderate-to-severe Crohn’s disease, relapsing-remitting sclerosis, and dry eye. However, there is no study on DN or a paradigmatic renal inflammatory disease, such as lupus nephritis. Taking into account the enormous wealth of information recently gathered about the role of inflammation in DN and that to be obtained in the future with the ongoing trials with anti-IL-17A antibodies in various clinical conditions, it is possible to envision the design of a similar trial in patients with moderate-to-severe progressive DN.

## Figures and Tables

**Figure 1 jcm-09-00272-f001:**
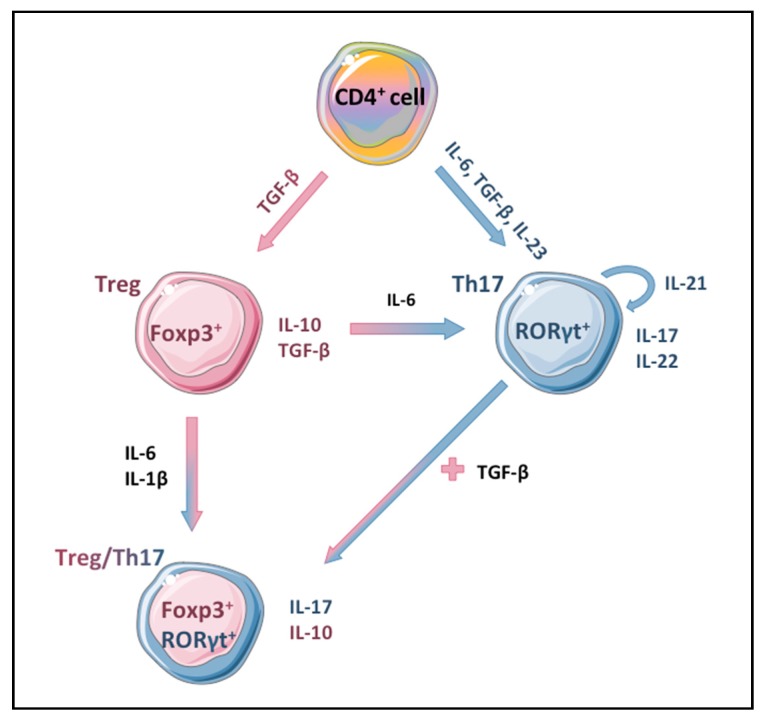
**Th17 differentiation.** Naïve CD4+ T lymphocytes can be differentiated into different T cell subtypes, including Treg or Th17 immune cells. This process is regulated by particular cytokines and activation of specific transcription factors, as indicated. Moreover, mixed phenotypes have also been described.

**Figure 2 jcm-09-00272-f002:**
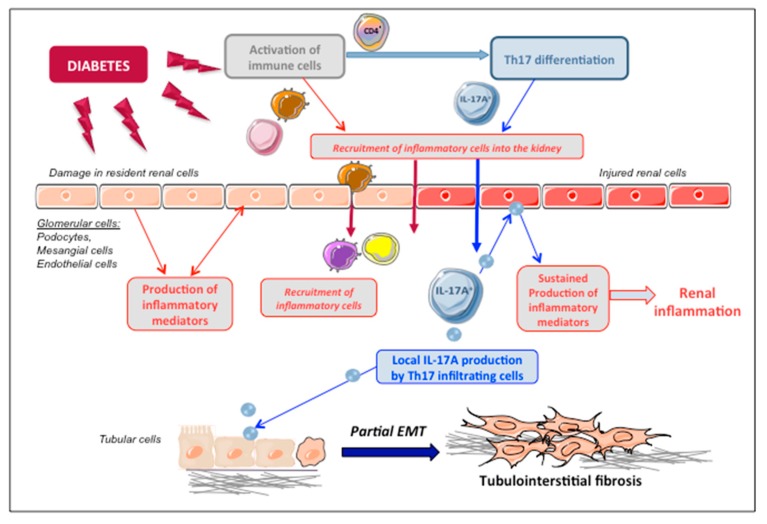
**Proposed mechanism of IL-17A-induced renal damage in diabetic nephropathy.** Under diabetic conditions, renal resident cells are activated and can produce different mediators that could contribute to recruit immune cells into the kidney. Infiltrating Th17 cells can locally produce IL-17A in the diabetic kidney. Then, IL-17A acting on IL-17R on resident renal cells can produce additional proinflammatory mediators, contributing to sustained inflammation. Moreover, IL-17A acting on tubular epithelial cells can induce phenotype changes, such as partial epithelial-to-mesenchymal transition (EMT) and secretome changes. By these mechanisms, IL-17A participates in the amplification of the inflammatory response and the progression of renal damage, finally leading to tubulointerstitial fibrosis.

**Figure 3 jcm-09-00272-f003:**
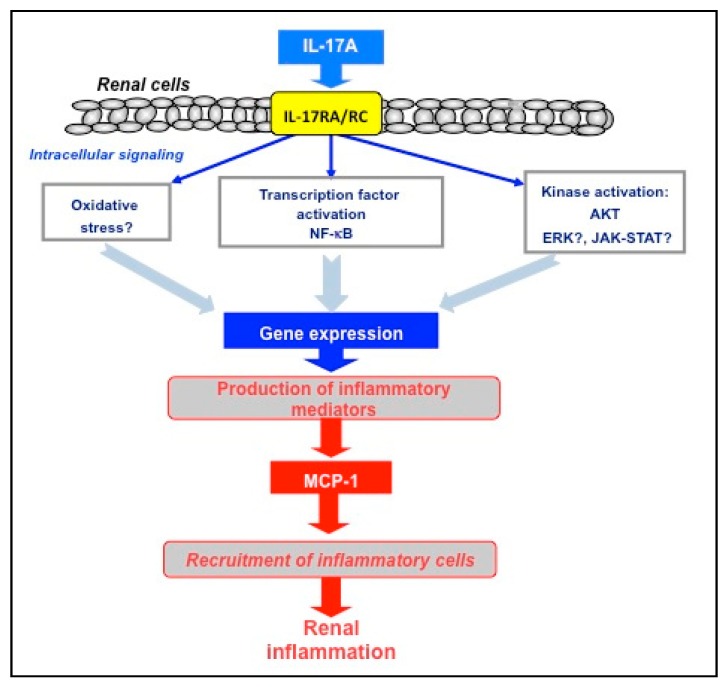
**Intracellular mechanisms involved in inflammatory responses of IL-17A in the kidney.** IL-17A can binds to its receptors and activates several intracellular mechanisms. The activation of NF-κB pathway and the upregulation of proinflammatory factors, such as MCP-1 can contribute to renal inflammation, as proposed under diabetic conditions. IL-17A can also activate other mechanisms, such as protein kinases and redox processes, but their role in renal damage have not been fully demonstrated.

**Table 1 jcm-09-00272-t001:** MicroRNAs involved in the regulation of Th17 differentiation.

Disease	microRNA	Targets	Reference
**Autoimmune diseases, including multiple sclerosis and its animal model, experimental autoimmune encephalomyelitis**	miR-20b	RORγt and STAT3	[118]
	miR-30a	IL-21R	[119]
	miR-146a	IL-6 and IL-21	[120]
	miR-106a-5p	RORC	[121]
	miR-214	mTOR signaling	[122]
	miR-9-5p	FOXP3	[121]
	miR-27a	TGFβ signaling	[122]
	miR-141 and miR-200a	SMAD2, GATA3 and FOXO3	[123]
	miR-155-3p	Dnaja2 and Dnajb2 (Hsp40)	[124]
	miR-17-92 cluster	PTEN and IKZF4	[125]
	miR-326	Ets-1	[126]
	miR-183-96-182 cluster	FOXO1	[127]
**Rheumatoid arthritis**	miR-363	Integrin αv/TGF-β	[128]
	miR-301a-3p	PIAS3	[129]
	miR-16	FOXP3	[130]
	miR-21	STAT5/FOXP3	[131]
**Ankylosing spondylitis**	miR-10b-5p	MAP3K7	[132]
**Psoriasis vulgaris**	miR-200a	FOXP3	[133]
	miR-210	FOXP3	[134]
**Systemic lupus erythematosus**	miR-873	Foxo1	[135]

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
