# Peer review of "Could IL-17A Be a Novel Therapeutic Target in Diabetic Nephropathy?"

_jcm, 2020, doi:10.3390/jcm9010272_

Round 1

Reviewer 1 Report

Dear Lavoz et al.,

       Thank you for submitting this manuscript. The influence of inflammation on the progression of Diabetic Nephropathy (DN) has become a growing target for DN treatments. I appreciate the description from the authors on current diabetic nephropathy treatments; it is unfortunate that although ACE inhibitors remain the standard of care, these drugs do not effectively reverse DN. 

Comments

The abstract needs some rewarding. For example, the sentence, "s. Over the time a number of patients progress to end-stage28 renal disease (ESRD) requiring dialysis and kidney transplantation." The sentence has an odd wording to it. The rest of the paper though as very good English grammar and wording. This is important for future readers to not be taken back by the wording in the abstract. Another oddly worded sentence is,"Currently, approximately 40%
29 of patients with diabetes developed kidney disease being the commonest cause of ESRD". I would suggest taking out the word "approximately". Have two adverbs in a row can make the sentence sound strange.  When discussing inflammation, I would also look into the literature on diabetic nephropathy. There have been clinical studies showing the presence of mast cells in the pathogenesis of DN. This would help support the focus of the paper on the importance of therapies directed at the inflammatory aspects of Diabetic Nephropathy.  I would suggest providing more molecular, biochemistry, and physiology of IL-17A in the manuscript. This will help the reader understand how IL-17A works in the cell and how it becomes disrupted in other inflammatory diseases.  I would also add a diagram showing the description of the clinical progression IL-17A in diabetic nephropathy. If there is not enough information, I would add more clinical data on the subject. I found a paper on pubmed you will want to check:https://www.ncbi.nlm.nih.gov/pubmed/27477820. The authors noted that after treatment of DN, there was a decrease in IL-17A. Specifically, the authors found that,"ost-treatment, there was a significant reduction in the CD4+CXCR5+PD-1+ Tfh cell counts and its subsets, with a corresponding decrease in plasma levels of IL-6 and IL-17A (p < 0.05) in DN patients, as compared to the HCs." I would suggest reading this paper. I think it will provide more clinical data and perhaps references to help support your argument further.  You should also examine this paper as well:https://www.ncbi.nlm.nih.gov/pubmed/28339909. This manuscript shows that IL-17 and CD40 interact together. Perhaps you could argue or suggest looking at whether targeting CD40 might be a better method for reducing IL-17 levels. In fact, this might be an argument against IL-17. Perhaps targeting CD40 might effectively reduce IL-17. But this would be important to discuss in the manuscript. It gives a fair analysis of the data. Interestingly, this paper found that,"Blockade with an anti-IL-17 monoclonal antibody reduced the expression of CD40 and TGF-β1, but increased the viability of cultured podocytes." This is a very important paper to have in your manuscript. It gives a good argument about the pathogensis and how blocking IL-17 can influence other receptors. Furthermore, this brings up important implications for cancer cells that express CD40 as shown in this paper, https://www.ncbi.nlm.nih.gov/pubmed/31412220.  I would also put lines in the table for microRNAs. It would help make the table easier to read.  In the manuscript, it would be important to emphasize the need for more clinical data. The data presented are mostly mouse models. The data is very strong but that's a potential drawback for therapeutic purposes. Many pharmaceuticals look promising but then fail during clinical trials.  I would also discuss the importance of IL-17A in non-proteinuric diabetic nephropathy. These patients progress of DN include increased vascular elements leading to hypertension. Here is a good review to cite and look through:https://www.ncbi.nlm.nih.gov/pubmed/31139314 This is also another good paper from scientific reports that argues for IL-17A in the progression of DN: https://www.ncbi.nlm.nih.gov/pubmed/30783187 Given the drugs targeting IL-17A, I would find out if IL-17A drugs have any benefits for other kidney diseases. This would help bolster the proof of concept for these drugs in DN.

       Overall, this was a very well-rewritten review. I myself did not know much about IL-17A. But given the clinical data, this does look like a promising area for research given the therapies approved by the FDA. I believe that if you add these papers, your paper will make a stronger argument and have the most recent clinical data to support further investigation. Thank you for giving me the opportunity to read through your manuscript. 

Reviewer 2 Report

General comment: The English requires attention and extensive correction. This issue if clear just from reading the abstract. I would recommend the authors seek help with language editing. The content of the paper is otherwise reasonable and the structure is easy to follow. 

I am not entirely sure Section 3 (biomarkers) adds much value to the paper. It seems like this section is quite generic and not necessary. 

I found the printed words in Figure 1 too small to read without enlarging to 150%. Is it possible to improve this? Furthermore, does the colour scheme have any meaning (red vs. blue)? This was not explained and there is no figure legend. A short legend would be quite useful. Ideally, the figure can be understood without referencing the main text.

I am not sure I can find any description of evidence to show that local (renal) levels of IL-17A is elevated in DN. Please correct me if I am wrong. The authors mention that circulating IL-17A levels decline rather than increase with progression of DN. I cannot find mention of proof that local IL-17A is higher in diabetic kidneys than non-diabetic kidneys to justify IL-17A inhibition in DN. This information would be nice to complete the story. Perhaps some images (immunohistochemistry or molecular quantification) would be informative, to show that Th17 cells and/or IL-17A is increased in DN.

Although the discussion of miRNAs is quite extensive (section 7), most of it is not relevant to IL-17 (section 7.2). Could it be more concisely presented?

A number of miRNAs in Table 1 have an asterisk but there is no footnote to explain it.

In Section 8, the authors discuss whether a therapeutic trial with anti-IL-17 mAB is feasible. However, they do not actually mention what barriers or issues exist (actual or theoretical) for using such a treatment for DN. Why do the authors think no one has embarked on such an investigation yet? What are the logistics, benefit-harm considerations, etc? What are the off-target effects of IL-17A inhibition? How would we gauge success in such trials or investigations? What work does the scientific community have to do before we get to the stage of clinical trials?
